# Inferring synaptic conductances from spike trains under a biophysically inspired point process model

**Kenneth W. Latimer**
The Institute for Neuroscience
The University of Texas at Austin
latimerk@utexas.edu

**E. J. Chichilnisky**
Department of Neurosurgery
Hansen Experimental Physics Laboratory
Stanford University
ej@stanford.edu

**Fred Rieke**
Department of Physiology and Biophysics
Howard Hughes Medical Institute
University of Washington
rieke@u.washington.edu

**Jonathan W. Pillow**
Princeton Neuroscience Institute
Department of Psychology
Princeton University
pillow@princeton.edu

## Abstract

A popular approach to neural characterization describes neural responses in terms of a cascade of linear and nonlinear stages: a linear filter to describe stimulus integration, followed by a nonlinear function to convert the filter output to spike rate. However, real neurons respond to stimuli in a manner that depends on the nonlinear integration of excitatory and inhibitory synaptic inputs. Here we introduce a biophysically inspired point process model that explicitly incorporates stimulus-induced changes in synaptic conductance in a dynamical model of neuronal membrane potential. Our work makes two important contributions. First, on a theoretical level, it offers a novel interpretation of the popular generalized linear model (GLM) for neural spike trains. We show that the classic GLM is a special case of our conductance-based model in which the stimulus linearly modulates excitatory and inhibitory conductances in an equal and opposite "push-pull" fashion. Our model can therefore be viewed as a direct extension of the GLM in which we relax these constraints; the resulting model can exhibit shunting as well as hyperpolarizing inhibition, and time-varying changes in both gain and membrane time constant. Second, on a practical level, we show that our model provides a tractable model of spike responses in early sensory neurons that is both more accurate and more interpretable than the GLM. Most importantly, we show that we can accurately infer intracellular synaptic conductances from extracellularly recorded spike trains. We validate these estimates using direct intracellular measurements of excitatory and inhibitory conductances in parasol retinal ganglion cells. The stimulus-dependence of both excitatory and inhibitory conductances can be well described by a linear-nonlinear cascade, with the filter driving inhibition exhibiting opposite sign and a slight delay relative to the filter driving excitation. We show that the model fit to extracellular spike trains can predict excitatory and inhibitory conductances elicited by novel stimuli with nearly the same accuracy as a model trained directly with intracellular conductances.

## 1 Introduction

The point process generalized linear model (GLM) has provided a useful and highly tractable tool for characterizing neural encoding in a variety of sensory, cognitive, and motor brain areas [1–5].

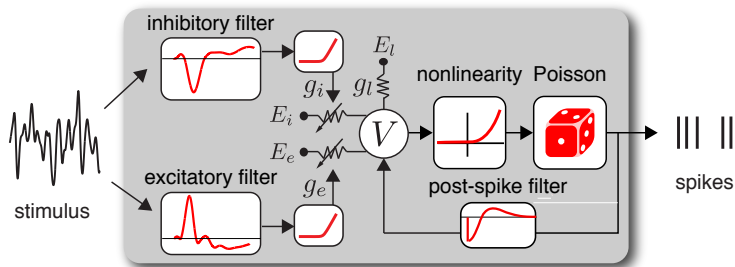

**Figure 1:** Schematic of conductance-based spiking model.

However, there is a substantial gap between descriptive statistical models like the GLM and more realistic, biophysically interpretable neural models. Cascade-type statistical models describe input to a neuron in terms of a set of linear (and sometimes nonlinear) filtering steps [6–11]. Real neurons, on the other hand, receive distinct excitatory and inhibitory synaptic inputs, which drive conductance changes that alter the nonlinear dynamics governing membrane potential. Previous work has shown that excitatory and inhibitory conductances in retina and other sensory areas can exhibit substantially different tuning. [12, 13].

Here we introduce a quasi-biophysical interpretation of the generalized linear model. The resulting interpretation reveals that the GLM can be viewed in terms of a highly constrained conductance-based model. We expand on this interpretation to construct a more flexible and more plausible conductance-based spiking model (CBSM), which allows for independent excitatory and inhibitory synaptic inputs. We show that the CBSM captures neural responses more accurately than the standard GLM, and allows us to accurately infer excitatory and inhibitory synaptic conductances from stimuli and extracellularly recorded spike trains.

## 2    A biophysical interpretation of the GLM

The generalized linear model (GLM) describes neural encoding in terms of a cascade of linear, nonlinear, and probabilistic spiking stages. A quasi-biological interpretation of GLM is known as "soft threshold" integrate-and-fire [14–17]. This interpretation regards the linear filter output as a membrane potential, and the nonlinear stage as a "soft threshold" function that governs how the probability of spiking increases with membrane potential, specifically:

$$
\begin{aligned}
V_t &= \mathbf{k}^\top \mathbf{x}_t && (1) \\
r_t &= f(V_t) && (2) \\
y_t | r_t &\sim \text{Poiss}(r_t \Delta_t), && (3)
\end{aligned}
$$

where $\mathbf{k}$ is a linear filter mapping the stimulus $\mathbf{x}_t$ to the membrane potential $V_t$ at time $t$, a fixed nonlinear function $f$ maps $V_t$ to the conditional intensity (or spike rate) $r_t$, and spike count $y_t$ is a Poisson random variable in a time bin of infinitesimal width $\Delta_t$. The log likelihood is

$$
\log p(y_{1:T} | \mathbf{x}_{1:T}, \mathbf{k}) = \sum_{t=1}^{T} -r_t \Delta_t + y_t \log(r_t \Delta_t) - \log(y_t!). \tag{4}
$$

The stimulus vector $\mathbf{x}_t$ can be augmented to include arbitrary covariates of the response such as the neuron's own spike history or spikes from other neurons [2, 3]. In such cases, the output does not form a Poisson process because spiking is history-dependent.

The nonlinearity $f$ is fixed *a priori*. Therefore, the only parameters are the coefficients of the filter $\mathbf{k}$. The most common choice is exponential, $f(z) = \exp(z)$, corresponding to the canonical 'log' link function for Poisson GLMs. Prior work [6] has shown that if $f$ grows at least linearly and at most exponentially, then the log-likelihood is jointly concave in model parameters $\theta$. This ensures that the log-likelihood has no non-global maxima, and gradient ascent methods are guaranteed to find the maximum likelihood estimate.

## 3 Interpreting the GLM as a conductance-based model

A more biophysical interpretation of the GLM can be obtained by considering a single-compartment neuron with linear membrane dynamics and conductance-based input:

$$\begin{aligned}
\frac{dV}{dt} &= -g_l V + g_e(t)(V - E_e) - g_i(t)(V - E_i) \\
&= -(g_l + g_e(t) + g_i(t))V + g_e(t)E_e + g_i(t)E_i \\
&= -g_{tot}(t)V + I_s(t),
\end{aligned} \tag{5}$$

where (for simplicity) we have set the leak current reversal potential to zero. The "total conductance" at time $t$ is $g_{tot}(t) = g_l + g_e(t) + g_i(t)$ and the "effective input current" is $I_s(t) = g_e(t)E_e + g_i(t)E_i$.

Suppose that the stimulus affects the neuron via the synaptic conductances $g_e$ and $g_i$. It is then natural to ask under which conditions, if any, the above model can correspond to a GLM. The definition of a GLM requires the solution $V(t)$ to be a linear (or affine) function of the stimulus. This arises if the two following conditions are met:

1. Total conductance $g_{tot}$ is constant. Thus, for some constant $c$:

$$g_e(t) + g_i(t) = c. \tag{6}$$

2. The input $I_s$ is linear in $x$. This holds if we set:

$$\begin{aligned}
g_e(\mathbf{x}_t) &= \mathbf{k}_e^\top \mathbf{x}_t + b_e \\
g_i(\mathbf{x}_t) &= \mathbf{k}_i^\top \mathbf{x}_t + b_i.
\end{aligned} \tag{7}$$

We can satisfy these two conditions by setting $\mathbf{k}_e = -\mathbf{k}_i$, so that the excitatory and inhibitory conductances are driven by equal and opposite linear projections of the stimulus. This allows us to rewrite the membrane equation (eq. 5):

$$\begin{aligned}
\frac{dV}{dt} &= -g_{tot}V + (\mathbf{k}_e^\top \mathbf{x}_t + b_e)E_e + (\mathbf{k}_i^\top \mathbf{x}_t + b_i)E_i \\
&= -g_{tot}V + \mathbf{k}_{tot}^\top \mathbf{x}_t + b_{tot},
\end{aligned} \tag{8}$$

where $g_{tot} = g_l + b_e + b_i$ is the (constant) total conductance, $\mathbf{k}_{tot} = \mathbf{k}_e E_e + \mathbf{k}_i E_i$, and $b_{tot} = b_e E_e + b_i E_i$. If we take the initial voltage $V_0$ to be $b_{tot}$, the equilibrium voltage in the absence of a stimulus, then the solution to this differential equation is

$$\begin{aligned}
V_t &= \int_0^t e^{-g_{tot}(t-s)} \left(\mathbf{k}_{tot}^\top \mathbf{x}_s\right) ds + b_{tot} \\
&= \mathbf{k}_{leak} * (\mathbf{k}_{tot}^\top \mathbf{x}_t) + b_{tot} \\
&= \mathbf{k}_{glm}^\top \mathbf{x}_t + b_{tot},
\end{aligned} \tag{9}$$

where $\mathbf{k}_{leak} * (\mathbf{k}_{tot}^\top \mathbf{x}_t)$ denotes linear convolution of the exponential decay "leak" filter $\mathbf{k}_{leak}(t) = e^{-g_{tot}t}$ with the linearly projected stimulus train, and $\mathbf{k}_{glm} = \mathbf{k}_{tot} * \mathbf{k}_{leak}$ is the "true" GLM filter (from eq. 1) that results from temporally convolving the conductance filter with the leak filter. Since the membrane potential is a linear (affine) function of the stimulus (as in eq. 1), the model is clearly a GLM.

Thus, to summarize, the GLM can be equated with a synaptic conductance-based dynamical model in which the GLM filter $\mathbf{k}$ results from a common linear filter driving excitatory and inhibitory synaptic conductances, blurred by convolution with an exponential leak filter determined by the total conductance.

## 4 Extending GLM to a nonlinear conductance-based model

From the above, it is easy to see how to create a more realistic conductance-based model of neural responses. Such a model would allow the stimulus tuning of excitation and inhibition to differ (i.e., allow $\mathbf{k}_e \neq -\mathbf{k}_i$), and would include a nonlinear relationship between $\mathbf{x}$ and the conductances to

preclude negative values (e.g., using a rectifying nonlinearity). As with the GLM, we assume that the only source of stochasticity on the model is in the spiking mechanism: we place no additional noise on the conductances or the voltage. This simplifying assumption allows us to perform efficient maximum likelihood inference using standard gradient ascent methods.

We specify the membrane potential of the conductance-based point process model as follows:

$$\frac{dV}{dt} = g_e(t)(E_e - V) + g_i(t)(E_i - V) + g_l(E_l - V), \tag{10}$$

$$g_e(t) = f_e(\mathbf{k}_e^\top \mathbf{x}_t), \qquad g_i(t) = f_i(\mathbf{k}_i^\top \mathbf{x}_t), \tag{11}$$

where $f_e$ and $f_i$ are nonlinear functions ensuring positivity of the synaptic conductances. In practice, we evaluate $V$ along a discrete lattice of points ($t = 1, 2, 3, \ldots T$) of width $\Delta_t$. Assuming $g_e$ and $g_i$ remain constant within each bin, the voltage equation becomes a simple linear differential equation with the solution

$$V(t+1) = e^{-g_{tot}(t)\Delta_t} \left( V(t) - \frac{I_s(t)}{g_{tot}(t)} \right) + \frac{I_s(t)}{g_{tot}(t)} \tag{12}$$

$$V(1) = E_l \tag{13}$$

$$g_{tot}(t) = g_e(t) + g_i(t) + g_l \tag{14}$$

$$I_s(t) = g_e(t)E_e + g_i(t)E_i + g_l E_l \tag{15}$$

The mapping from membrane potential to spiking is similar to that in the standard GLM (eq. 3):

$$r_t = f(V(t)) \tag{16}$$

$$f(V) = \exp\left( \frac{(V - V_T)}{V_S} \right) \tag{17}$$

$$y_t | r_t \sim \text{Poiss}(r_t \Delta_t). \tag{18}$$

The voltage-to-spike rate nonlinearity $f$ follows the form proposed by Mensi et al. [17], where $V_T$ is a soft spiking threshold and $V_S$ determines the steepness of the nonlinearity. To account for refractory periods or other spike-dependent behaviors, we simply augment the function to include a GLM-like spike history term:

$$f(V) = \exp\left( \frac{(V - V_T)}{V_S} + \mathbf{h}^\top \mathbf{y}^{hist} \right) \tag{19}$$

Spiking activity in real neurons influences both the membrane potential and the output nonlinearity. We could include additional conductance terms that depend on either stimuli or spike history, such as an after hyper-polarization current; this provides one direction for future work. For spatial stimuli, the model can include a set of spatially distinct rectified inputs (e.g., as employed in [9]).

To complete the model, we must select a form for the conductance nonlinearities $f_e$ and $f_i$. Although we could attempt to fit these functions (e.g., as in [9, 18]), we fixed them to be the soft-rectifying function:

$$f_e(\cdot), f_i(\cdot) = \log(1 + \exp(\cdot)). \tag{20}$$

Fixing these nonlinearities improved the speed and robustness of maximum likelihood parameter fitting. Moreover, we examined intracellularly recorded conductances and found that the nonlinear mapping from linearly projected stimuli to conductance was well described by this function (see Fig. 4).

The model parameters we estimate are $\{\mathbf{k}_e, \mathbf{k}_i, b_e, b_i, \mathbf{h}, g_l, E_l\}$. We set the remaining model parameters to biologically plausible values: $V_T = -70mV, V_S = 4mV, E_e = 0mV$, and $E_i = -80mV$. To limit the total number of parameters, we fit the linear filters $\mathbf{k}_e$ and $\mathbf{k}_i$ using a basis consisting of 12 raised cosine functions, and we used 10 raised cosine functions for the spike history filter [3].

The log-likelihood function for this model is not concave in the model parameters, which increases the importance to selecting a good initialization point. We initialized the parameters by fitting a simplified model which had only one conductance. We initialized the leak terms as $E_l = -70mV$ and $g_l = 200$. We assumed a single synaptic conductance with a linear stimulus dependence, $g_{lin}(t) = \mathbf{k}_{lin}^\top \mathbf{x}_t$ (note that this allows for negative conductance values). We initialized this filter

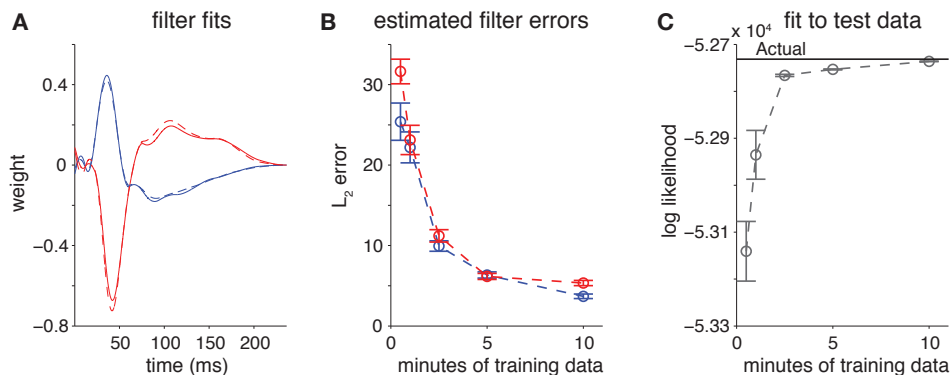

**Figure 2:** Simulation results. **(A)** Estimates (solid traces) of excitatory (blue) and inhibitory (red) stimulus filters from 10 minutes of simulated data. (Dashed lines indicate true filters). **(B)** The $L_2$ norm between the estimated input filters and the true filters (calculated in the low-dimensional basis) as a function of the amount of training data. **(C)** The log-likelihood of the fit CBSM on withheld test data converges to the log likelihood of the true model.

the GLM fit, and then numerically maximized the likelihood for $\mathbf{k}_{lin}$. We then initialized the parameters for the complete model using $\mathbf{k}_e = c\mathbf{k}_{lin}$ and $\mathbf{k}_i = -c\mathbf{k}_{lin}$, where $0 < c \leq 1$, thereby exploiting the mapping between the GLM and the CBSM. Although this initialization presumes that excitation and inhibition have nearly opposite tuning, we found that standard optimization methods successfully converged to the true model parameters even when $\mathbf{k}_e$ and $\mathbf{k}_i$ had similar tuning (simulation results not shown).

## 5 Results: simulations

To examine the estimation performance, we fit spike train data simulated from a CBSM with known parameters (see Fig. 2). The simulated data qualitatively mimicked experimental datasets, with input filters selected to reproduce the stimulus tuning of macaque ON parasol RGCs. The stimulus consisted of a one dimensional white noise signal, binned at a 0.1ms resolution, and filtered with a low pass filter with a 60Hz cutoff frequency. The simulated cell produced a firing rate of approximately 32spikes/s. We validated our maximum likelihood fitting procedure by examining error in the fitted parameters, and evaluating the log-likelihood on a held out five-minute test set. With increasing amounts of training data, the parameter estimates converged to the true parameters, despite the fact that the model does not have the concavity guarantees of the standard GLM.

To explore the CBSM's qualitative response properties, we performed simulated experiments using stimuli with varying statistics (see Fig. 3). We simulated spike responses from a CBSM with fixed parameters to stimuli with different standard deviations. We then separately fit responses from each simulation with a standard GLM. The fitted GLM filters exhibit shifts in both peak height and position for stimuli with different variance. This suggests that the CBSM can exhibit gain control effects that cannot be captured by a classic GLM with a spike history filter and exponential nonlinearity.

## 6 Results: neural data

We fit the CBSM to spike trains recorded from 7 macaque ON parasol RGCs [12]. The spike trains were obtained by cell attached recordings in response to full-field, white noise stimuli (identical to the simulations above). Either 30 or 40 trials were recorded from each cell, using 10 unique 6 second stimuli. After the spike trains were recorded, voltage clamp recordings were used to measure the excitatory and inhibitory conductances to the same stimuli. We fit the model using the spike trains for 9 of the stimuli, and the remaining trials were used to test model fit. Thus, the models were effectively trained using 3 or 4 repeats of 54 seconds of full-field noise stimulus. We compared the intracellular recordings to the $g_e$ and $g_i$ estimated from the CBSM (Fig. 5). Additionally, we fit the measured conductances with the linear-nonlinear cascade model from the CBSM (the terms $g_e$ and

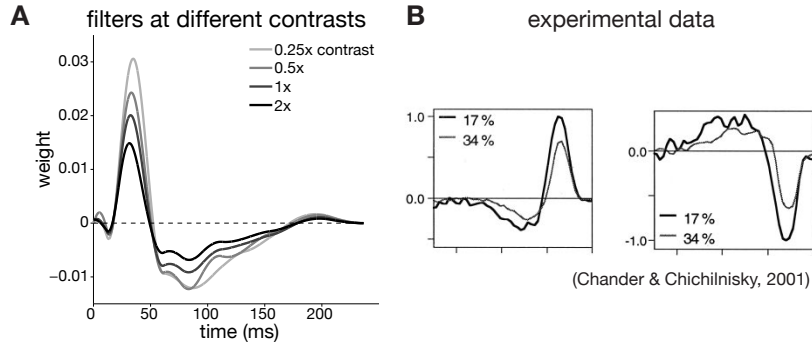

**Figure 3:** Qualitative illustration of model's capacity to exhibit contrast adaptation (or gain control). **(A)** The GLM filters fit to a fixed CBSM simulated at various levels of stimulus variance. **(B)** Filters fit to two real retinal ganglion cells at two different levels of contrast (from [19]).

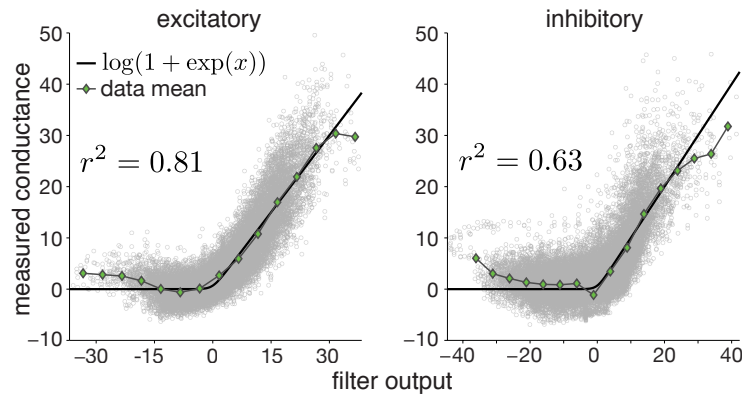

**Figure 4:** Measured conductance vs. output of a fitted linear stimulus filter (gray points), for both the excitatory (left) and inhibitory (right) conductances. The green diamonds correspond to a non-parametric estimate of the conductance nonlinearity, given by the mean conductance for each bin of filter output. For both conductances, the function is is well described by a soft-rectifying function (black trace).

$g_i$ in eq. 11) with a least-squares fit as an upper bound measure for the best possible conductance estimate given our model. The CBSM correctly determined the stimulus tuning for excitation and inhibition for these cells: inhibition is oppositely tuned and slightly delayed from excitation.

For the side-by-side comparison shown in Fig. 5, we introduced a scaling factor in the estimated conductances in order to compare the conductances estimated from spike trains against recorded conductances. Real membrane voltage dynamics depend on the capacitance of the membrane, which we do not include because it introduces an arbitrary scaling factor that cannot be estimated by spike alone. Therefore, for comparisons we chose a scaling factor for each cell independently. However, we used a single scaling for the inhibitory and excitatory conductances. Additionally, we often had 2 or 3 repeated trials of the withheld stimulus, and we compared the model prediction to the average conductance recorded for the stimulus. The CBSM predicted the synaptic conductances with an average $r^2 = 0.54$ for the excitatory and an $r^2 = 0.39$ for the inhibitory input from spike trains, compared to an average $r^2 = 0.72$ and $r^2 = 0.59$ for the excitatory and inhibitory conductances respectively from the least-squares fit directly to the conductances (Fig. 6). To summarize, using only a few minutes of spiking data, the CBSM could account for $71\%$ of the variance of the excitatory input and $62\%$ of the inhibitory input that can possibly be explained using the LN cascade model of the conductances (eq. 11).

One challenge we discovered when fitting the model to real spike trains was that one filter, typically $\mathbf{k}_i$, would often become much larger than the other filter. This resulted in one conductance becoming dominant, which the intracellular recordings indicated was not the case. This was likely due to the fact that we are data-limited when dealing with intracellular recordings: the spike train recordings include only 1 minute of unique stimulus. To alleviate this problem, we added a penalty term, $\phi$, to

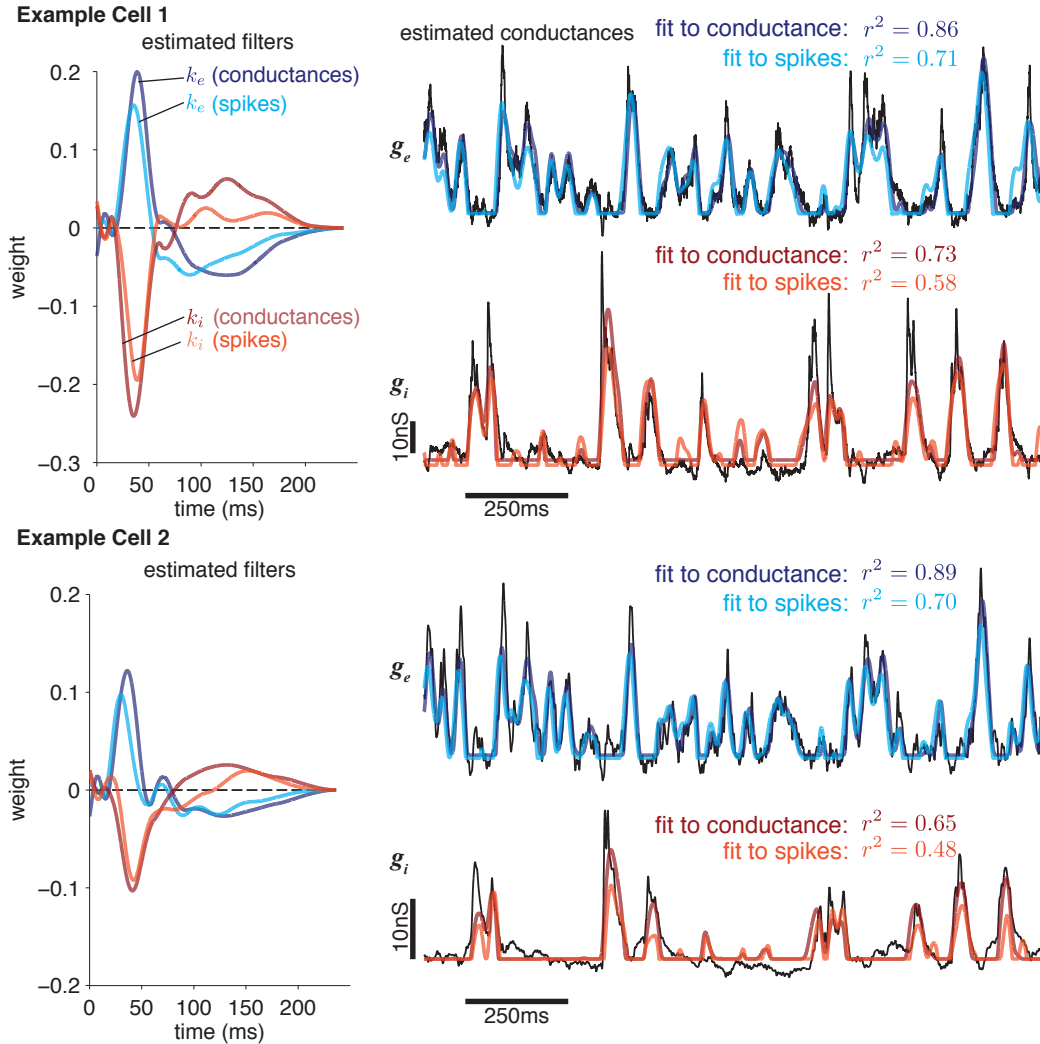

**Figure 5:** Two example ON parasol RGC responses to a full-field noise stimulus fit with the CBSM. The model parameters were fit to spike train data, and then used to predict excitatory and inhibitory synaptic currents recorded separately in response to novel stimuli. For comparison, we show predictions of an LN model fit directly to the conductance data. **Left**: Linear kernels for the excitatory (blue) and inhibitory (red) inputs estimated from the conductance-based model (light red, light blue) and estimated by fitting a linear-nonlinear model directly to the measured conductances (dark red, dark blue). The filters represent a combination of events that occur in the retinal circuitry in response to a visual stimulus, and are primarily shaped by the cone transduction process. **Right**: Conductances predicted by our model on a withheld test stimulus. Measured conductances (black) are compared to the predictions from the CBSM filters (fit to spiking data) and an LN model (fit to conductance data).

the log likelihood on the difference of the $L_2$ norms of $\mathbf{k}_e$ and $\mathbf{k}_i$:

$$\phi(\mathbf{k}_e, \mathbf{k}_i) = \lambda \left( ||\mathbf{k}_e||^2 - ||\mathbf{k}_i||^2 \right)^2 \tag{21}$$

This differentiable penalty ensures that the model will not rely too strongly on one filter over the other, without imposing any prior on the shape of the filters (with $\lambda = 0.05$). We note that unlike the a typical situation with statistical models that contain more abstract parameters, the terms we wish to regularize can be measured with intracellular recordings. Future work with this model could include more informative, data-driven priors on $\mathbf{k}_e$ and $\mathbf{k}_i$.

Finally, we fit the CBSM and GLM to a population of nine extracellularly recorded macaque RGCs in response to a full-field binary noise stimulus [20]. We used a five minute segment for model fitting, and compared predicted spike rate using a 6s test stimulus for which we had repeated trials.

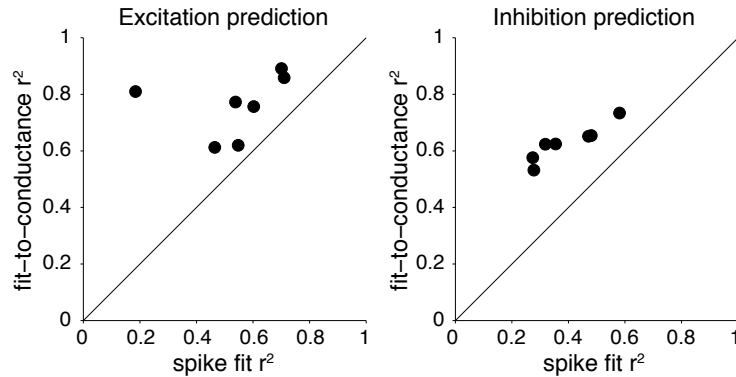

**Figure 6:** Summary of the CBSM fits to 7 ON parasol RGCs for which we had both spike train and conductance recordings. The axes show model's ability to predict the excitatory (left) and inhibitory (right) inputs to a new stimulus in terms of $r^2$. The CBSM fit is compared against predictions of an LN model fit directly to measured conductances.

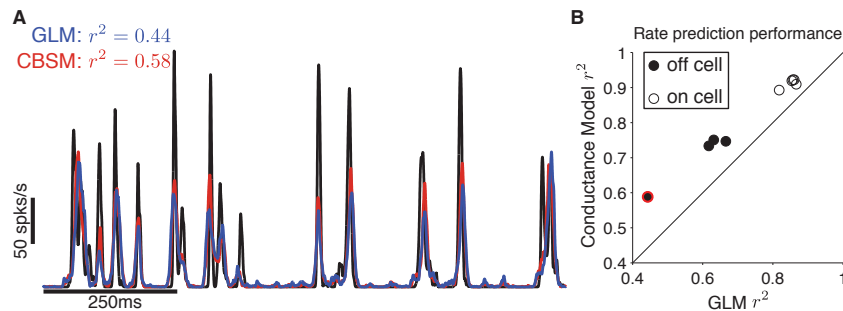

**Figure 7: (A)** Performance on spike rate (PSTH) prediction. The true rate (black) was estimated using 167 repeat trials. The GLM prediction is in blue and the CBSM is in red. The PSTHs were smoothed with a Gaussian kernel with a 1ms standard deviation. (B) Spike rate prediction performance for the population of 9 cells. The red circle indicates cell used in left plot.

The CBSM achieved a 0.08 higher average $r^2$ in PSTH prediction performance compared to the GLM. All nine cells showed an improved fit with the CBSM.

# 7 Discussion

The classic GLM is a valuable tool for describing the relationship between stimuli and spike responses. However, the GLM describes this map as a mathematically convenient linear-nonlinear cascade, which does not take account of the biophysical properties of neural processing. Here we have shown that the GLM may be interpreted as a biophysically inspired, but highly constrained, synaptic conductance-based model. We proposed a more realistic model of the conductance, removing the artificial constraints present in the GLM interpretation, which results in a new, more accurate and more flexible conductance-based point process model for neural responses. Even without the benefit of a concave log-likelihood, numerical optimization methods provide accurate estimates of model parameters.

Qualitatively, the CBSM has a stimulus-dependent time constant, which allows it change gain as a function of stimulus statistics (e.g., contrast), an effect that cannot be captured by a classic GLM. The model also allows the excitatory and inhibitory conductances to be distinct functions of the sensory stimulus, as is expected in real neurons. We demonstrate that the CBSM not only achieves improved performance as a phenomenological model of neural encoding compared to the GLM, the model accurately estimates the tuning of the excitatory and inhibitory synaptic inputs to RGCs purely from measured spike times. As we move towards more naturalistic stimulus conditions, we believe that the conductance-based approach will become a valuable tool for understanding the neural code in sensory systems.

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
