[Reviews · NeurIPS 2014]

Submitted by Assigned_Reviewer_8

The authors propose a conductance based spiking model (CBSM) that is more biophysically
realistic than the currently popular generalized linear model (GLM). Furthermore, the authors
present CBSM as a generalization of the GLM and propose a set of constraints that can reduce it
to a GLM and a GLM variant that would be as adaptive as the CBSM.
The proposed model is an interesting extension to current spiking models in that it is parametrized
in a more descriptive way of the spiking process without sacrificing much of the mathematical
convenience of the GLM.
One thing that could raise some concerns stems from the last paragraph of page 6. The behaviour of
the inhibitory filter described in line 322 is reminiscent of overfitting. The authors deal with that
by adding the regularizing penalty of Eq. 22. However, it seems that they don't use cross-validation
in the experimental results to show the applicability of their model in a more natural environment.

Some small things that might need correction:

line 121: fourth subscript should be i?
line 214: chose our an?
Summary: Good quality paper in line that is a considerable contribution to the state of the art. I would like to see a few more experiments

Submitted by Assigned_Reviewer_12

Well written and significant paper on inferring synaptic conductance from recorded spike trains, with validation from physiological data.

How is this method extended to infer network connectivity structure, in addition to the synaptic conductances, from spike data on a limited number of neurons in the networks? Please see Abarbanel et al and other recent work for some examples in this direction.
Summary: Well written and significant paper on inferring synaptic conductance from recorded spike trains, with validation from physiological data.

Submitted by Assigned_Reviewer_33

The authors improved a state space method of inferring the synaptic conductance. The method was originally proposed as a GLM based on a linear assumption in ref [8]. The point of improvement in this contribution is to incorporate the nonlinear stimulus-induced changes in synaptic conductance in a dynamical model of neuronal membrane potential. The proposed method offers an interpretation of GLM in terms of conductance-based model, and makes it possible to infer the synaptic conductance and time series of inputs. The proposed method appears solid and I believe this paper deserves publication, provided that they could revise the ms responding the following comment.

Major comment

The temporal profiles of filters used in the simulation have time-scales of 30-50 ms, which appear very long compared to the commonly known timescales of the biological conductance. I would request the authors to argue about the reason of this choice, or they should redo the analysis with the smaller timescales.

Minor typos

p3: "g_e(t) E_i" should be "g_i(t) E_i"
p7 "this this"
Summary: Improvement of a state space method of inferring synaptic conductance, offering an interpretation of GLM in terms of conductance-based model. It would make a significant contribution if the problem of timescale is discussed properly.
Author Feedback
Author rebuttal: We thank the reviewers for their thoughtful comments. We greatly appreciate the time and effort of all reviewers, and believe their comments will improve the clarity of our manuscript.

Reviewer #1:
----------
Issue: extending the framework to estimate network connectivity.

Thank you for your review. We have not explored any extensions for this model to infer network connectivity; we have only applied our methods to modeling stimulus-conductance relationships. In the RGCs we examine, much of the presynaptic input comes from cells that do not produce action potentials, and are therefore not observed in multi-electrode population recordings. Therefore, estimating connectivity (at least, beyond functional connectivity estimated with GLM-style linear coupling filters), while an fascinating and challenging suggestion for future studies, is beyond the scope of our paper.

Reviewer #2:
----------
Issue: the temporal profiles of the conductance filters appear much longer than typical biological time constants.

While this is true for typical synapses, the filters represent a transformation of luminance to the total conductance as measured at the soma instead of a single release of neurotransmitter in a presynaptic cell. The filter therefore represents the combination of many events that occur in the retinal circuitry in response to a visual stimulus, and are primarily shaped by the cone transduction process. The filters we used were in fact estimated from intracellular recordings. We thank the reviewer for pointing out this area of potential confusion, and we will clarify our definition of conductance filter in our writing.

Reviewer #3:
----------
Thank you for your enthusiastic review.

Issue: over-fitting of inhibitory filter.

We agree with the reviewer's assessment that the behavior of our estimator for the inhibitory filters in the model likely shows signs of overfitting. We did use cross validation for evaluating the model performance (fitting to 90% of the data and reporting performance on a held out 10% of the data). Moreover, the limited amount of the data from these recordings (60 seconds of a unique stimulus segment, with 3 to 4 repeats) gave us worries about holding out a second fold of data for setting hyper-parameters governing regularization. However, we agree with the reviewer that this is an important issue, and in future work we will more carefully explore methods for regularizing the filter estimates using either cross-validation or Bayesian based methods.